# Prediction of Drug-Drug Interaction Using an Attention-Based Graph Neural Network on Drug Molecular Graphs

**DOI:** 10.3390/molecules27093004

**Published:** 2022-05-07

**Authors:** Yue-Hua Feng, Shao-Wu Zhang

**Affiliations:** MOE Key Laboratory of Information Fusion Technology, School of Automation, Northwestern Polytechnical University, Xi’an 710072, China; feng_yuehua@mail.nwpu.edu.cn

**Keywords:** drug-drug interaction, prediction, feature representation, molecular graph, graph attention network

## Abstract

The treatment of complex diseases by using multiple drugs has become popular. However, drug-drug interactions (DDI) may give rise to the risk of unanticipated adverse effects and even unknown toxicity. Therefore, for polypharmacy safety it is crucial to identify DDIs and explore their underlying mechanisms. The detection of DDI in the wet lab is expensive and time-consuming, due to the need for experimental research over a large volume of drug combinations. Although many computational methods have been developed to predict DDIs, most of these are incapable of predicting potential DDIs between drugs within the DDI network and new drugs from outside the DDI network. In addition, they are not designed to explore the underlying mechanisms of DDIs and lack interpretative capacity. Thus, here we propose a novel method of GNN-DDI to predict potential DDIs by constructing a five-layer graph attention network to identify *k*-hops low-dimensional feature representations for each drug from its chemical molecular graph, concatenating all identified features of each drug pair, and inputting them into a MLP predictor to obtain the final DDI prediction score. The experimental results demonstrate that our GNN-DDI is suitable for each of two DDI predicting scenarios, namely the potential DDIs among known drugs in the DDI network and those between drugs within the DDI network and new drugs from outside DDI network. The case study indicates that our method can explore the specific drug substructures that lead to the potential DDIs, which helps to improve interpretability and discover the underlying interaction mechanisms of drug pairs.

## 1. Introduction

Polypharmacy, also termed drug combination treatment, has become a promising strategy for treating complex diseases (e.g., diabetes and cancer) in recent years [1]. For example, Pembrolizumab has been combined with Sorafenib in the treatment of metastatic hepatocellular carcinoma [2]. Entacapone increases the plasma concentration of Levodopa and improves therapeutic effects on Parkinson’s disease [3]. Nevertheless, the combined use of two or more drugs (i.e., drug-drug interactions, DDIs) triggers pharmacological changes that may result in unexpected effects (e.g., side effects, adverse reactions, and even serious toxicity) [4]. As the need for polypharmacy treatments increases, identification of DDIs has become urgent. Nevertheless, it is expensive and time-consuming to detect DDIs among drug pairs on a large scale both in vitro and in vivo. To screen DDIs, computational approaches, especially machine learning-based methods, have been developed to deduce potential drug-drug interactions [5].

Existing computational approaches can be roughly classified into three categories: text-mining-based, machine-learning-based, and deep-learning-based methods. Textmining-based approaches discover and collect recorded DDIs from the scientific literature, electronic medical records [6,7], insurance claim databases, and the FDA Adverse Event Reporting System. They use Natural Language Processing (NLP) technology to extract DDI information from various formats of text, and they are very useful in building DDI-related databases [7,8,9,10,11,12,13]. However, these approaches are incapable of detecting unrecorded DDIs, and they cannot give an alert to potential drug interactions before drug combination treatment [14]. 

With the advantages of both high efficiency and low costs, various machine learning methods have been shown promise in providing preliminary screening of DDIs for further experimental validation. Generally, models are trained by using confirmed DDIs to infer the potential DDIs among massive quantities of unlabeled drug pairs. The training involves diverse drug properties, such as chemical structure [14,15,16,17,18], targets [14,15,18,19], anatomical taxonomy [16,19,20], and phenotypic observation [17,18,19,20]. The models transform the DDI prediction task that infers whether or not a drug interacts with another into a binary classification problem. These methods are usually implemented according to established classifiers (e.g., KNN [16], SVM [16], logistic regression [14,20], decision tree [21], and naïve Bayes [21]), network propagation of reasoning behind drug-drug network structures [20,22], label propagation [23], random walk [15] and probabilistic soft logic [19,21], or matrix factorization [17,18,24]. Generally, traditional machine learning methods rely heavily on the quality of handcrafted features derived from the drug properties.

In terms of extracting features from data without manual input [25], deep learning methods, especially graph convolution network methods, provide promising routes into the field of drug development and discovery [5], such as molecular activity prediction, drug side effect prediction [17] drug target interactions prediction [25], drug response [26,27,28,29], and drug synergy [30,31,32,33,34]. Those methods in the field of drug-drug interaction prediction contribute to traditional binary DDI prediction [35] or multi-type DDI prediction [36,37,38,39]. Some of these methods have constructed deep learning frameworks to learn latent features from various properties of drugs, and other methods have built models to extract the latent features from the DDI network [40], including the homogeneous DDI network and the heterogeneous knowledge network [41]. For example, NDD [42] calculated the corresponding drug similarity matrix from several drug properties, and inputted it into a multi-layer deep learning classifier for predicting binary DDIs. Wang et al. [41] extracted drug representation features by utilizing GCN from the DDI networks, and inputted them into a three-layer multilayer perception (MLP) for predicting binary DDIs. KGNN [43] constructed a drug knowledge graph that includes various entities such as drug target, side effect, and pathway disease, and used the graph representation method to extract drug features from this huge heterogeneous graph to predict DDIs. The methods [36,37,38] first treat the rows in a drug similarity matrix as corresponding drug feature vectors, and set the concatenation of two feature vectors as the feature vector to represent a pair of drugs, and then train a multi-layer DNN with feature vectors and types of DDIs as the classifiers to predict multi-type DDIs.

Although these methods achieved inspiring results, they had several limitations as follows. First, those methods extracting the latent features from the DDI network relied on the network’s topological information, thus they are blind to new drugs that have no links with the drugs in the DDI network. Secondly, current deep learning methods lack interpretation of drug interactions, and it is difficult to observe the underlying mechanisms of drug interactions. To address these issues, here we propose a novel GNN-DDI method to predict drug-drug interaction. GNN-DDI constructed a five-layer graph attention network to identify *k*-hops low-dimensional feature representations for each drug from its chemical molecular graph, and then concatenates the learned features for each drug pair, inputting them into a MLP predictor to obtain the final DDI prediction score. The multi-layer GAT of CSGN-DDI can capture different kth-order substructure functional groups of the drug molecular graph through multi-step operations, to generate effective feature representation of the drugs. The experimental results demonstrate that our GNN-DDI is superior in predicting the potential DDIs between the drugs within the DDI network and new drugs outside the DDI network. GNN-DDI helps to improve interpretability and reveal the underlying mechanisms of drug pair interactions.

## 2. Materials and Methods 

### 2.1. Datasets

We first built the DDI dataset that contains 1,935 drugs and 589,827 annotated drug-drug interactions from DrugBank 5.0 [44]. Then we downloaded the completed XML-formatted database (including the comprehensive profiles of 11,440 drugs), and parsed all approved small-molecule drugs and their DDI entries. We extracted the drugs’ chemical structure information using Simplified Molecular Input Line Entry System (SMILES) strings from the XML file provided by DrugBank, and transformed them into the corresponding molecular structure graph using the open-source library RDKit (Figure 1). These drug molecular graphs were taken as the input graphs for the graph convolutional network in the feature extractor of GNN-DDI to obtain the drug feature vectors. In each molecular graph, atoms were denoted as nodes, edges representing the bond between atoms, and each node containing a 78-dim initial feature vector including the symbol of the atom (i.e., 44-dimension, one-hot code), the number of adjacent atoms, the implied valence of the atom, its formal charge, the number of free radical electrons, the hybridization of the atom (i.e., 5-dimension, one hot code), the number of hydrogen bonds, and whether the atom is aromatic.

GNN-DDI learns the drug representation features directly from their chemical molecular structure graphs by graph convolution network. In order to compare those features with other molecular structure fingerprint features and features from their biological properties, we also extracted the ATC (Anatomical Therapeutic Chemical Classification) and DBP (Drug Binding Proteins) from DrugBank, and utilized the PubChem fingerprint and the MACCSkeys fingerprint (Molecular ACCess System keys fingerprint [45] to convert the SMILES of drugs into the 881-dimesion and 166-dimension binary vector, respectively. Each bit in the vector indicates the occurrence or non-occurrence of a pre-defined substructure according to Pubchem fingerprints or MACCSkeys fingerprints. ATC codes are released by the World Health Organization [46], and they categorize drug substances at different levels according to organs they affect, application area, therapeutic properties, chemical, and pharmacological properties. It is generally accepted that compounds with similar physicochemical properties exhibit similar biological activity. To feed the 7-bit ATC code into a deep learning model, we converted the data into a one-hot code with 118 bits. We also used drug-binding protein (DBP) data [47], containing 899 drug targets and 222 non-target proteins. Similarly, each drug was represented as a binary DBP-based feature vector, with each bit indicating whether the drug binds to a specific protein.

### 2.2. Problem Formulation

Let G be (n+m) drugs including *n* known drugs $G_{1}=\left\{d_{i}\right\}$ and *m* new drugs $G_{2}=\left\{d_{j}\right\}$, where $G_{1}\cup G_{2}=G$ and $G_{1}\cap G_{2}=\emptyset$, and $D_{1}=\left\{G_{x},G_{y}\right\}s$ is the interaction between $G_{x}\epsilon G_{1}$ and $G_{y}\epsilon G_{1}$, and $D_{2}=\left\{G_{x},G_{z}\right\}$ is the interaction between $G_{x}\epsilon G_{1}$ and $G_{z}\epsilon G_{2}$. In addition, each drug can be represented as a molecular structure graph, and we denote it by a graph $G_{i}\left(V_{i},E_{i}\right)$, where $V_{i}=\left\{V_{i1},V_{i2},\ldots ,V_{ip}\right\}$ is the set of nodes representing the atoms in the drug $d_{i}$, $E_{i}=\left\{{\left(V_{is},\;V_{it}\right)}_{s,t=1}^{p}\right\}$ is the set of edges representing the bonds connecting two atoms in the drug $d_{i}$, and ${H_{i}^{\left(0\right)}=(h_{i1}^{\left(0\right)},h_{i2}^{\left(0\right)},\ldots ,h_{ip}^{\left(0\right)})}^{T}$ is the initial feature matrix of p nodes in $G_{i}$ of drug $d_{i}$. Our task is to deduce DDI candidates among those unannotated drug-drug pairs based on known DDIs. There are two different scenarios of DDI prediction as follows: 

The first prediction task is to learn a function mapping $ℱ:G_{1}\times G_{1}\to \left\{0,1\right\}$ to deduce the potential interactions among the unlabeled pairs of drugs in $G_{1}$ (Figure 2A).

The second prediction task is to learn a function mapping $ℱ:G_{1}\times G_{2}\to \left\{0,1\right\}$ to deduce the potential interactions among the unlabeled drug pairs between $G_{1}$ and $G_{2}$ (Figure 2B). We used all known DDIs $\left\{G_{x},G_{y}\right\}\in D_{1}|G_{x}\in G_{1}\;and\;G_{y}\in G_{1}$ to train the prediction model for predicting all unlabeled drug pairs $\left\{G_{x},G_{y}\right\}\in D_{2}|G_{x}\in G_{1}\;and\;G_{y}\in G_{2}$.

### 2.3. GNN-DDI Model

In this work, we propose a representation learning framework, GNN-DDI, to predict drug-drug interactions. GNN-DDI mainly consists of two modules: a drug feature extractor and a DDI predictor (Figure 3). The first module is composed of a five-layer graph attention convolutional network (GAT) [48] that learns the function $f_{e}\left(G_{i}\right)$ to obtain the latent feature vector $Z_{i}$ of each drug from its molecular structure graph $G_{i}\left(V_{i},E_{i}\right)$, where $Z_{i}\in R^{1\times k}$. The latent vectors ($Z_{i}$ and $Z_{j}$) of two drugs are concatenated to form the feature vector $Z_{ij}$ of the corresponding drug pair. In each layer of the feature extractor, the convolutional operation aggregates information from its atomic neighborhood and updates the node feature for each atomic node in a drug molecular structure graph. Through several convolutional layers, informative features of drug chemical functional groups within its whole chemical structure are captured, that are critical in drug interactions. The second module is a multi-layer perception that predicts the probability score of drug pair interaction by taking the feature vector $Z_{ij}$ of the drug pair as the input. The overall algorithm of GNN-DDI is shown in Algorithm 1.

Algorithms 1 The pseudo-code of GNN-DDI.
**Algorithms 1** The pseudo-code of GNN-DDIinput: Molecular graph Gx of drug x and its original features Hi(0) of atomic nodes               Molecular graph Gy of drug y and its original features Hi(0) of atomic nodes output: Probability score p(Gx,Gy) of drug pair (x,y)1: Initialize parameter sets in GNN-DDI.2: for k in K:3: Compute hx(k+1) and hy(k+1) based on Equations (1) to (3).4: SAGPooling based on Equation (4) to obtain HGx(k+1) and Hy(k+1) in layer k.5: end for6: Concatenate k-hops HGx(k+1) and Hy(k+1) based on Equation (5) to obtain HGx and HGy.7: Concatenate HGx and HGy to obtain the latent feature vector of a drug pair H(Gx,Gy)8: Feed feature vector H(Gx,Gy) into the predictor to get probability score p(Gx,Gy).

#### 2.3.1. Feature Extractor

Each drug has its molecular structure graph, in which atoms are denoted as nodes, and edges represent the bonds between atoms. Because the numbers of atoms and chemical bonds in the molecular graph of each drug are different, each molecular graph can be learned by the graph convolutional network to generate drug informative representation. The graph convolutional network consists of an information aggregation function and an update function. The former continuously gathers neighborhood information for each node in the graph, and the latter updates the gathered information to obtain the informative representation features for each node. 

The traditional convolution network aggregates the neighborhood information of each node in the molecular graph without difference. Due to the different importance of neighbor nodes, the weighted aggregation can obtain more effective representations for drugs and be conductive to disclosing drug interaction mechanisms. Therefore, we designed a five-layer graph convolutional network with an attention mechanism [48,49] to generate the embedding representation for each atomic node in the drug molecular graph. Each node is represented as a latent feature vector, which contains the information about its neighborhood in the drug molecular graph without manual feature engineering. 

(A) Information aggregation and update

In each layer of the feature extractor, the convolutional operation aggregated information by weighting from its atomic neighborhood and updated the node feature for each atomic node in a drug molecular structure graph. Through several convolutional layers, we captured informative features of drug chemical functional groups within each drug’s whole chemical structure, that are critical in drug interactions. 

For any layer in the GNN-DDI feature extractor (Figure 2A), the general propagation rule is defined as:(1)hi(k+1)=σ(∑j∈NiαijW(k)hj(k)+W(k)hi(k))
where Ni denotes the set of atomic node neighbors in Gx, hi(k) is the input feature vector, hi(0) is the original features of each atomic node in molecular graph (details in Section 2.1), W(k) is the trainable weight matrix in the *k*-th layer of Gx, σ is a non-linear element-wise activation function (i.e., ReLU), and αij denotes the aggregation weight between the updating node vxi and its neighborhood node vxj determining the relevant importance between them. αij can be calculated by the attention mechanism as follows:(2)αij=softmax(eij)=exp(eij)∑k∈Niexp(eik)
(3)eij=LeakyReLU(a→T[watthi(k)∥watthj(k))
where a→T∈ℝ2F′ is a shared weight vector composed of a layer of feedforward neural network, T  is a transpose operation, LeakyReLU is an activation function [50], and ∥ denotes the concatenated operation.

(B) Pooling of atomic feature vectors

The feature extractor takes the molecular structure graph and atomic original features of each drug as input, to output the latent feature vector Z of each drug using a multi-layer graph convolution network. In each layer, the neighborhood information of each atomic node vxi in drug molecular graph Gx is continuously aggregated to update the feature hi(k) of node vxi, hence an updated feature vector matrix Hi(k)∈ℝp×k of each atom in drug Gx is obtained, here *p* is the number of atoms in drug Gx and k is the dimension of this layer. The feature matrix is taken as input to the next layer of the feature extractor module. To predict interactions among drug pairs, the feature matrix Hi(k) of the drug molecular graph must be transformed into the drug feature vector HGi(k). Therefore, after convolutional operations in each layer, we adopted SAGPooling [49] to implement this transform operation:(4)HGi(k)=∑inγihi(k)               
where γi is the feature weight of each atomic node vxi in the whole molecular graph Gx, which represents the importance of each node in the molecular graph. γi is determined according to the topological and contextual information of node vxi by SAGPooling. 

As the learned representation features are drawn from different multi-head attention in different subspaces, the multi-head attention mechanism can improve the model’s learning stability and enhance its expression ability [51]. Therefore, we adopted multi-head attention in the feature extractor. Assuming *L* heads are adopted, in each layer of the feature extractor, there are *L* information aggregation and update operations from Equations (1)–(3) in parallel, and *L* same dimension representation features of each node are obtained. Then they are concatenated together as the final feature $h_{i}^{\left(k\right)}$.

#### 2.3.2. Feature Aggregation for Drug Pairs 

So far, five *k*-hop latent feature vectors $H_{G_{x}}^{\left(k\right)}$ of each drug were obtained from five-layer GAT. Different *k*-hops of feature vectors involve various neighbor receptive fields, therefore they contain various sizes of sub-structures in a drug molecular graph. For example, the molecular chemical structure graphs of two drugs Hydroquinone (DrugBank ID: DB09526) and Acetic acid (DrugBank ID: DB03166) are shown in Figure 4 respectively. They are both weak acids due to the sub-structures of phenolic hydroxyl AROH and carboxyl COOH. 

In order to correctly extract the sub-structure ArOH in hydroquinone, we need a three-hop information aggregation from the neighborhoods in its molecular graph. In the same way, we only need a two-hop convolution operation to correctly extract the sub-structures COOH from Acetic acid. However, traditional graph representation networks usually use a fixed-sized receptive field (i.e., using the final feature vectors from the last layer of graph convolution network for downstream tasks), which may result in either incomplete sub-structures being extracted (i.e., receptive fields are too small), or redundant sub-structures being included (i.e., receptive fields are too large). In order to solve this limitation, all five *k*-hop latent feature vectors $H_{G_{x}}^{\left(k\right)}$ of each drug were concatenated as the final representation feature of the drug for the downstream prediction task.
(5)$$Z_{G_{x}}={∥}_{k=1}^{K}H_{G_{x}}^{\left(k\right)}$$
where ∥ denotes the concatenated operation.

Finally, we concatenated the latent feature vectors of two drugs in each drug pair to form a feature vector $h\left(G_{x},G_{y}\right)=[Z_{G_{x}},Z_{G_{y}}]$ to represent the drug pair, and took $h\left(G_{x},G_{y}\right)$ as the input of MLP to predict the probability value of interaction between two drugs.

#### 2.3.3. MLP Predictor

GNN-DDI converts the DDI prediction task into a binary classification problem. Because MLP has been proved to give excellent performance in classification, we constructed a five-layer MLP as the predictor (Figure 3). ReLU was selected as the activation function in the first four layers, while the activation function SoftMax was selected in the last layer, which maps the output score into the range of 0–1, representing how likely potential DDIs are in drug pairs. 

In the GNN-DDI training process, the binary cross-entropy loss function was adopted to continuously optimize the model.
(6)$$L\left(p,q\right)=-\sum_{i,j}y_{ij}\;\log\left(p\left(G_{x},G_{y}\right)\right)+\left(1-y_{ij}\right)(1-\log\left(p\left(G_{x},G_{y}\right)\right)$$
where $y_{ij}$ is the true label (i.e., 0 or 1) of the training drug pair $\left(G_{x},G_{y}\right)$, $p\left(G_{x},G_{y}\right)$ is the predicting probability value generated by the MLP predictor. Through continuous reduction of the loss function, the model is optimized.

### 2.4. Cross-Validation Strategy and Assessment Metrics

In order to evaluate the performance of GNN-DDI, we employed two different cross-validation strategies of sample set partition. The first one is the edge set partition strategy, in which all interaction edges were randomly partitioned into 80% training edges (which includes 5% validation edges) and 20% test edges. The other one is the drug partition strategy, in which all drugs were randomly partitioned into 80% training drugs and 20% test drugs. As shown in Figure 5A, the edge set A was the training set and the edge set B was the test set in the edge partition strategy. However, in the drug partition strategy, as the interactions between drugs in the training set and in the test set were deleted, the drugs in the test set are regarded as new drugs. Meanwhile, those new drugs did not appear in the training process, which was completely new to the model. Therefore, the interactions among training drugs were taken as the training samples, and the interactions between new drugs and training drugs as the test samples. For example, as shown in Figure 5B, drugs d1 to d5 were the training drugs and d6 to d8 were the test drugs. All edges (in set A set) between the training drugs were used as the training samples, and all edges (in set B) between the training drugs and the test drugs were used as the test samples. The drug partition strategy can measure the performance of a predictor when new drugs appear. All the strategies were repeated 10 times, and the average results were used to evaluate the prediction performance of GNN-DDI. 

Accuracy (ACC), precision, recall, F1 score, AUC (i.e., area under the receiver operating characteristic curve), and AUPR (i.e., area under the precision-recall curve) were used to assess the performance of GNN-DDI. The receiver operating characteristic curve reveals the relationship between true-positive rate (precision) and false-positive rate based on various thresholds. The precision-recall curve reveals the relationship between precision (true-positive rate) and recall based on various thresholds. These metrics are defined as follows:(7)$$\mathrm{Accuracy}=\frac{TP+TN}{TP+FP+TN+FN}$$
(8)$$\mathrm{Precision}=\frac{TP}{TP+FP}$$
(9)$$\mathrm{Recall}=\frac{TP}{TP+FN}$$
(10)$$F_{1}=\frac{2\times Precision\times Recall}{Precision+Recall}$$
where TP, FP, TN, and FN refer to the numbers of true positive samples, false positive samples, true negative samples, and false negative samples, respectively.

## 3. Results and Discussion

In this section, we first introduce the GNN-DDI hyper-parameters, then compare the performance of GNN-DDI with other existing methods in both DDI prediction scenarios. We also demonstrate the effectiveness of structural features learned by using the feature extractor in GNN-DDI. Finally, through a case study we investigate the respective substructures of a drug pair leading to a potential DDI. 

### 3.1. Parameter Setting

To learn an optimal model of DDI prediction, we first determined the architecture of GNN-DDI. The model consisted of 5 layers of attention-mechanism-based graph convolution network in which each layer had 2 attention heads. The feature dimension of each head was 32-dimension (32-dim), so the total feature dimension of each layer was 64-dim. The drug feature dimension outputted from the feature extractor was 320 (64 × 5), thus the number of neurons in the input layer of the MLP predictor was 640 (i.e., the dimension of a drug pair). The dimension of the other three hidden layers was determined empirically. The numbers of neurons in each of the three hidden layers were 128, 64, and 32, respectively.

With this feature extractor architecture and the MLP predictor, we performed a grid search with an Adam optimizer [52] to tune the hyper-parameters (i.e., epoch, learning rate, and batch size) of GNN-DDI. The epoch (i.e., the number of training iterations) was tuned from the list of values {20, 60,100, 200, 400, 600, 1000}. The learning rate (determining whether and when the objective function converges to the optimal values) was empirically investigated from the list {0.0001, 0.001, 0.005, 0.01, 0.05, 0.1}. The mini-batch strategy (i.e., sampling a fixed number of drug pairs in each batch) was tuned from the list {50, 200, 400, 600, 1000, 2000}. We finally experimentally determined a well-trained GNN-DDI by setting the epoch at 400, the learning rate at 0.001, and the batch size at 1024.

### 3.2. Results of GNN-DDI and Five Other Methods in the First Prediction Scenario

To validate the performance of GNN-DDI in the first prediction scenario (i.e., predicting the interactions of drugs within the DDI network), we compared our GNN-DDI method with other five state-of-the-art methods: two of Vilar’s methods (named as Vilar 1 and Vilar 2, respectively) [53,54], the label propagation-based method (LP) [23], Zhang’s method [15] and DPDDI [22]. Vilar 1 [53] identified potential DDIs by integrating a Tanimoto similarity matrix of molecular structures with the known DDI matrix through a linear matrix transformation. Vilar 2 [54] used drug interaction profile fingerprints (IPFs) to measure similarity for predicting DDIs. The LP method [23] applied label propagation to assign labels from known DDIs to previously unlabeled nodes by computing drug-similarity-derived weights of edges within the DDI network. Zhang’s method [15] collected a variety of drug-related data (e.g., known drug-drug interactions, drug substructures, targets, enzymes, transporters, pathways, indications, and side effects) to build 29 base classifiers (i.e., KNN, random walk, matrix disturbed method, etc.), then developed a classifier ensemble model to predict DDIs. DPDDI [22] constructed a graph convolution network to learn the network structure features of drugs from the DDI network for predicting potential drug interactions within the DDI network. In this section, all comparing methods used the edge partition strategy to split the DDI edges into training edges and test edges. 

The comparison results of GNN-DDI against the five other methods are shown in Table 1, from which we can see that GNN-DDI achieved the best results. It outperformed four other state-of-the-art methods in terms of AUPR, Recall, Precision and F_1_. GNN-DDI achieved improvements of 8.5~22.9%, 8.9~66.8%, 13.2~42.5%, 9.4~57%, and 11.8~53.5% against the Vilar 1, Vilar 2, LP, and Zhang methods in terms of AUPR, recall, precision, ACC, and F1 score, respectively. 

Although the AUC of our GNN-DDI was little lower than that of DPDDI and Zhang’s method, and the ACC of our GNN-DDI was lower than that of DPDDI, the performance results in terms of AUPR, recall, precision, and F_1_ for GNN-DDI are higher than that for DPDDI and Zhang’s method. Zhang’s method used nine drug-related data sources, while GNN-DDI used only the drug molecular graph. More importantly, Zhang’s method and DPDDI can only work in the first DDI prediction scenario, that is, they predict only the interactions between known drugs, and cannot predict the interactions between known drugs and new drugs (i.e., the second DDI prediction scenario). 

### 3.3. Results of GNN-DDI and Four Other Methods in the Second Prediction Scenario

In this section, we evaluated the performance of GNN-DDI in the second DDI prediction scenario (i.e., predicting the interactions between known drugs and new drugs) by using the drug partition strategy to split the drugs in the DDI network into the training drugs and testing drugs. The new drugs did not appear in the training process. Therefore, the drug partition strategy is able to measure the performance of prediction methods when new drugs appear. We compared our GNN-DDI method with other four different chemical- and biological-feature-based prediction methods. These four compared methods include two chemical-structure feature-based methods (the PubChem feature-based method and the MACCSkeys feature-based method), the ATC feature-based method, and the DBP feature-based method. The DBP method extracted 3334-dim structure features, and the ATC method extracted 118-dim structure features. The PubChem feature-based method extracted 881-dim features from the PubChem fingerprint, and the MACCSkeys feature-based method extracted 166-dim features from the MACCSkeys fingerprint. These molecular structure features derived from GNN-DDI, MACCSkeys, PubChem, DBP and ATC feature descriptions of drugs were respectively concatenated to feed the MLP predictor of GNN-DDI for DDI prediction. Figure 6 shows the AUCs and ACCs of GNN-DDI and four other methods in the second DDI prediction scenario, from which we can see that GNN-DDI achieved the best results. 

### 3.4. Effects of Using Different Feature Extraction Approaches

The GNN-DDI feature extractor consists of a five-layer GAT network to learn the latent feature vectors of drugs. In each layer, the convolutional operation aggregates information from its atomic neighborhood and updates the node feature for each atomic node in a drug molecular structure graph. Through several convolutional layers, we captured the informative features of drug chemical functional groups within the whole chemical structure. In order to evaluate the effectiveness of the molecular structure features learned by GNN-DDI, we compared these with two structure features derived from the PubChem fingerprint (named the PubChem feature) and MACCSkeys fingerprint feature (named the MACCSkeys feature), and the drug’s chemical and biological features according to DBP and ATC. These features were respectively concatenated to feed the MLP predictor of GNN-DDI for DDI prediction.

The comparison results are shown in Table 2, from which we can see that the structure feature learned by the GNN-DDI feature extractor outperformed the other four features in terms of AUC, AUPR and recall. Specifically, the structure feature learned by GNN-DDI achieved improvements of 0.6~4.8%, 0.2~5.5%, 0.4~11.7% against the other four features from the PubChem fingerprint, MACCSkeys fingerprint, DBP, and ATC in terms of AUC, AUPR, recall, respectively. Although the precision, ACC, and F_1_ of the structure feature learned by GNN-DDI are lower than those of the PubChem feature and MACCSkeys feature, the structure features extracted directly from the drug molecular graph by the five-layer GAT network in GNN-DDI can explore the specific substructures of drugs. This can improve interpretability and reveal the underlying mechanisms of drug pair interactions.

### 3.5. Interpretability Case Studies

In this section, we will explore the specific substructures of drugs that result in potential DDIs. Different layers of graph convolutional network involved various neighbor receptive fields of the drug molecular graph. The five *k*-hop latent feature vectors $H_{G_{x}}^{\left(k\right)}$ of each drug contained various sizes of sub-structures in its molecular graph. We reserved these *k*-hop latent feature vectors ${\left\{H_{G_{x}}^{\left(k\right)}\right\}}_{k=1}^{K}$ and ${\left\{H_{G_{y}}^{\left(k\right)}\right\}}_{k=1}^{K}$ of each drug pair from different layers of the feature extractor in GNN-DDI, and selected the two features with the largest scores as the most contributing substructure features to the potential interaction of this drug pair.
(11)$$Score=MAX\left(H_{G_{x}}^{\left(k\right)}\;\cdot \;H_{G_{y}}^{\left(k\right)}{}^{T}\right),\;i=1,2,\ldots ,K;j=1,2,\ldots ,K$$
where K=5 and “ … ” denotes the inner product operation. The larger the inner product value, the greater the contribution of substructure features to the potential interaction of this drug pair. The *k*-hop latent feature vector $H_{G_{x}}^{\left(k\right)}$ was derived from the atomic feature matrix $H_{i}^{\left(k\right)}$ in the drug molecular graph $G_{x}$ by SAGPooling [49] (Equation (4)). The feature weight of each atomic node $\gamma_{i}$ in the pooling process represents the importance of each node in the molecular graph, and is determined according to the topological and contextual information of the node in the drug molecular graph $G_{x}$ by SAGPooling. According to the atomic weight $\gamma_{i}$ in feature vectors $H_{G_{x}}^{\left(s\right)}$ and $H_{G_{y}}^{\left(t\right)}$, we drew the weighted molecular structure graphs of two drugs $G_{x}$ and $G_{y}$ to illustrate the specific substructures that contribute to potential interaction of drug pair ($G_{x}$,$G_{y}$), and help to discover the underlying mechanisms of DDIs. 

We selected three interactions between Sildenafil and other nitrate-based drugs (Isosorbide mononitrate, Nitroglycerin, Amyl Nitrite) as a case study [55]. Sildenafil is an effective treatment for erectile dysfunction and pulmonary hypertension [56]. Sildenafil was developed as a phosphodiesterase-5 (PDE5) inhibitor. In the presence of a PDE5 inhibitor, nitrate (NOO3)-based drugs such as Isosorbide mononitrate can cause dramatic increases in cyclic guanosine monophosphate [57] (Murad 1986), which leads to intense lowering of blood pressure that can cause heart attacks [58]. 

We drew the heat map of the weighted molecular structure graphs for each drug pair according to the atomic weight $\gamma_{i}$ in feature vectors $H_{G_{x}}^{\left(s\right)}$ and $H_{G_{y}}^{\left(t\right)}$ (Figure 6). Each row in Figure 7 contains a pair of drugs and the descriptions of corresponding interactions. In the heat map, the important contributing substructures are mainly concentrated near its center (represented by green circles). From the heat map, we can see that the specific substructure of the nitrate group (NOO3) contributes highly to the interaction between Sildenafil and other nitrate-based drugs (Isosorbide Mononitrate, Nitroglycerin, Amyl Nitrite).

## 4. Conclusions

Aiming to address the problem that current DDI prediction methods are incapable of predicting potential interactions for new drugs and always lack interpretability, we proposed a novel method GNN-DDI to predict potential DDIs by constructing a five-layer graph attention network (GAT) to learn k-hops low-dimensional feature representations of each drug from its chemical molecular graph. The learned features of each drug pair were concatenated, and fed into an MLP to output the final DDI prediction score. The multi-layer GAT of GNN-DDI can capture different kth-order substructure functional groups of the drug molecular graph through multi-step operations, to generate the effective feature representation of drugs. The experimental results demonstrate that GNN-DDI achieved superior performance in each of two DDI predicting scenarios, namely potential DDIs among known drugs and between known drugs and new drugs. In addition, the performance of drug features directly learned by GNN-DDI from drug chemical molecular graphs is better than that obtained from drug chemical structure fingerprints, biological features and ATC features, which proves the feature effectiveness derived from our method. In the case study we selected three interactions between Sildenafil and other nitrate-based drugs, which lead to intense lowering of blood pressure that can cause heart attacks. More importantly, the result shows that our GNN-DDI can explore specific drug substructures that can result in potential DDIs, helping to improve interpretability and to discover the underlying interaction mechanisms of drug pairs.

## Figures and Tables

**Figure 1 molecules-27-03004-f001:**
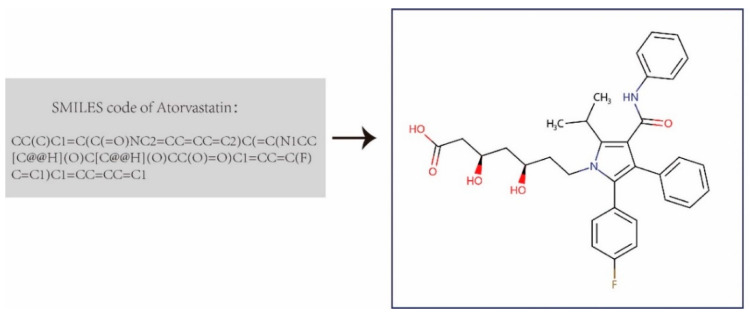
Drug molecular graph transformed from drug SMILES.

**Figure 2 molecules-27-03004-f002:**
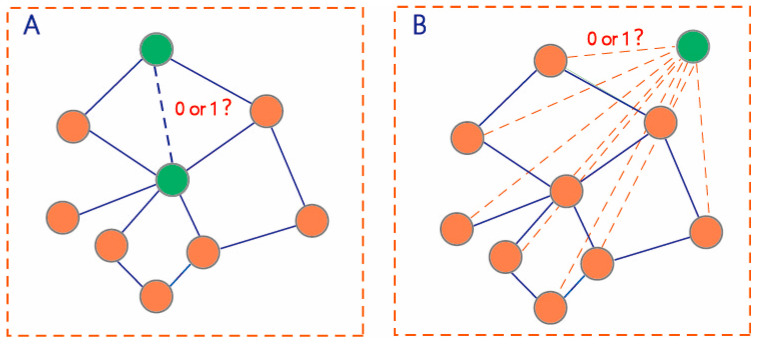
Two scenarios of DDI prediction. (**A**) DDI prediction among drugs in the DDI network (**B**) DDI prediction between the drugs within the DDI network and new drugs outside the network.

**Figure 3 molecules-27-03004-f003:**
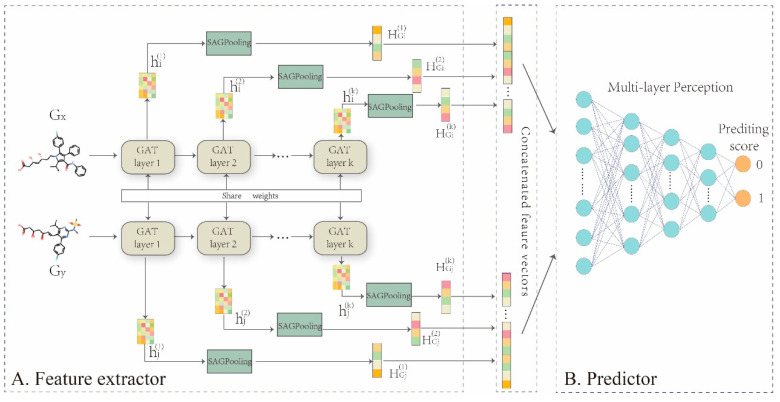
Overall framework of GNN-DDI. (**A**) drug feature extractor. The five-layer GAT networks are built to encode the molecular structure graph of each drug into its feature vectors $H_{i}^{\left(k\right)}={\left(h_{i1}^{\left(k\right)},{\;h}_{i2}^{\left(k\right)},\ldots ,h_{\mathrm{ip}}^{\left(k\right)}\right)}^{T}$, to capture topological properties especially chemical functional groups within the whole chemical structure graph, which are critical in drug interactions. In addition, the atomic nodes feature vectors $H_{i}^{\left(k\right)}$ in the molecular graph output from each layer are transformed to drug feature $H_{G_{i}}^{\left(k\right)}$ by SAGPooling in each layer, and those drug features $H_{G_{i}}^{\left(k\right)}$ are concatenated together as final drug feature vector $Z_{G_{i}}$. (**B**) DDI predictor. Concatenating two drug latent features $Z_{G_{i}}$ and $Z_{G_{j}}$ to feed into a MLP for implementing the prediction task.

**Figure 4 molecules-27-03004-f004:**
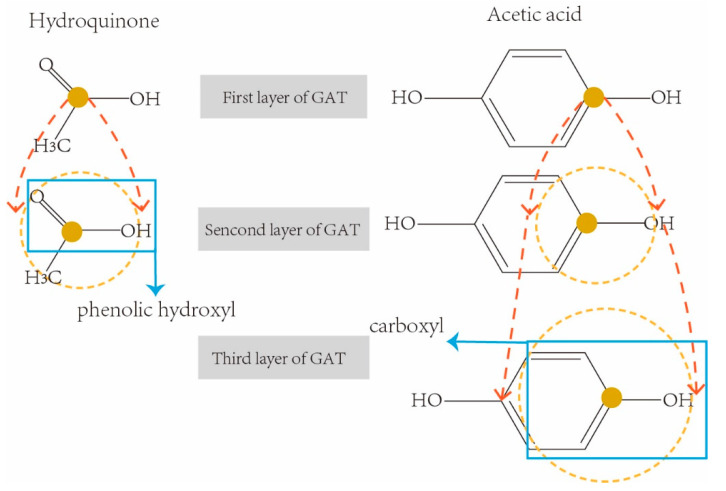
Examples of receptive fields with *k*-hop convolution network.

**Figure 5 molecules-27-03004-f005:**
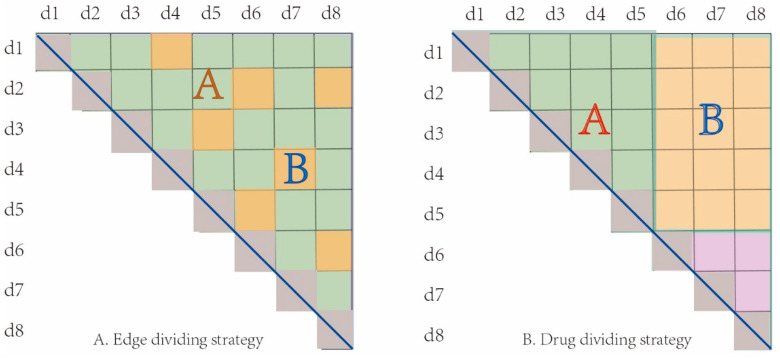
Two cross-validation strategies of sample partitioning. (**A**) Edge partition strategy, (**B**) Drug partition strategy.

**Figure 6 molecules-27-03004-f006:**
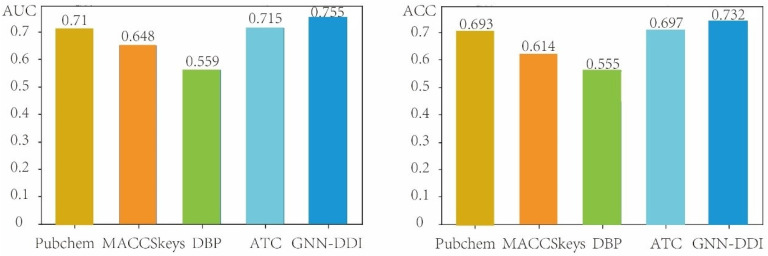
Comparison results of GNN-DDI with four other methods in the second DDI prediction scenario.

**Figure 7 molecules-27-03004-f007:**
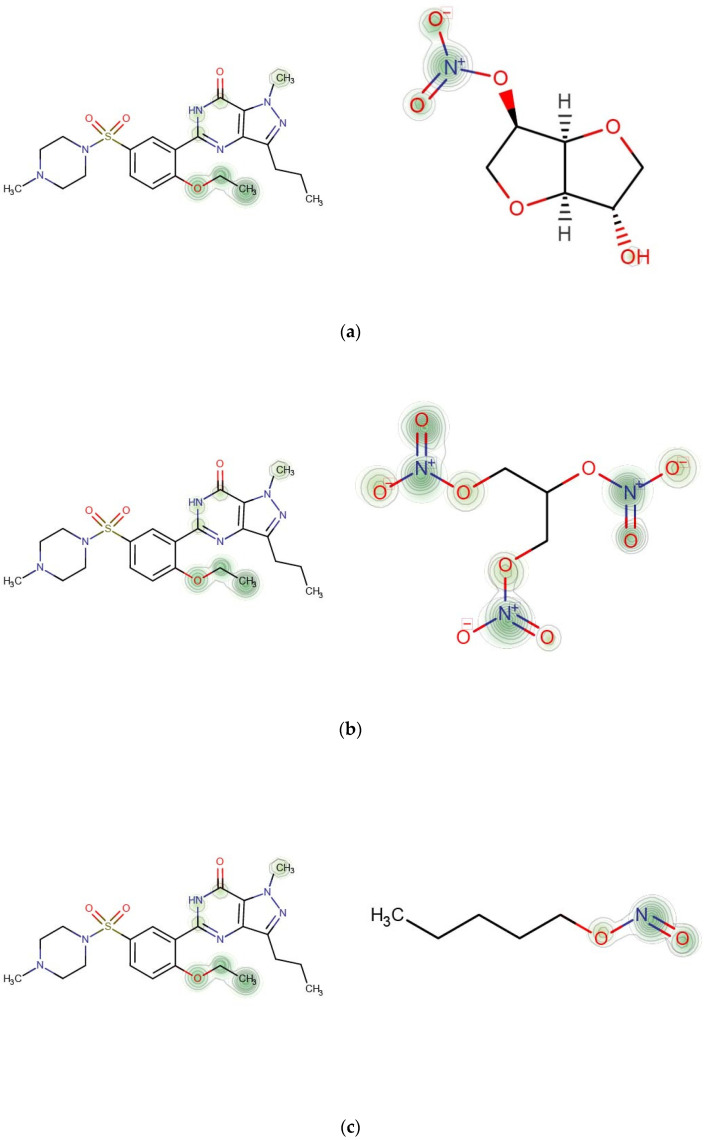
Contributions of specific substructures to drug interactions. (**a**) Sildenafil (k = 4) and Isosorbide Mononitrate (k = 3). Description in DrugBank: The risk or severity of hypotension can be increased when Isosorbide mononitrate is combined with Sildenafil. (**b**) Sildenafil (k = 4) and Nitroglycerin (k = 3). Description in DrugBank: The risk or severity of hypotension can be increased when Nitroglycerin is combined with Sildenafil. (**c**) Sildenafil (k = 4) and Amyl Nitrite (k = 3). Description in DrugBank: The risk or severity of hypotension can be increased when Amyl Nitrite is combined with Sildenafil.

**Table 1 molecules-27-03004-t001:** Results of GNN-DDI and other five methods in the first prediction scenario.

Methods	AUC	AUPR	Recall	Precision	ACC	*F* _1_
Vilar 1 [53]	0.707	0.262	0.495	0.253	0.719	0.334
Vilar2 [54]	0.826	0.533	0.569	0.515	0.862	0.540
LP [23]	0.851	0.799	0.685	0.729	0.809	0.706
Zhang [15]	0.954	0.841	0.788	0.717	0.934	0.751
DPDDI	0.956	0.907	0.810	0.754	0.940	0.840
GNN-DDI	0.936	0.930	0.920	0.823	0.863	0.869

**Table 2 molecules-27-03004-t002:** Comparison results of the structure features learned by GNN-DDI and other chemical and biological features.

	AUC	AUPR	Recall	Precision	ACC	F_1_
Pubchem features	0.920	0.928	0.880	0.862	0.905	0.883
MACCSkeys features	0.930	0.924	0.879	0.864	0.901	0.882
DBP features	0.862	0.875	0.803	0.757	0.89	0.819
ATC features	0.888	0.895	0.834	0.811	0.871	0.840
GNN-DDI features	0.936	0.930	0.920	0.823	0.861	0.869

## Data Availability

The source code and associated datasets used in this work are publicly available at https://github.com/NWPU-903PR/GNN-DDI (accessed on 7 April 2022).

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
