# Peer review of "Prediction of Drug-Drug Interaction Using an Attention-Based Graph Neural Network on Drug Molecular Graphs"

_molecules, 2022, doi:10.3390/molecules27093004_

Round 1

Reviewer 1 Report

General comment:

The computer science aspect of this manuscript is interesting to follow, as it offers fresh computational insight toward DDI problem, and depicting specialized algorithmic design for this issue. However, as this is a life sciences journal, not computer science one, It is clear that the biological foundation of this approach should be improved in order to be deserved for publication.

Specific comments:

  1. Line 28-37: Better add more extensive references on polypharmacy and drug-drug interaction to strengthen the biological foundation. Please you can cite some more references from PUBMED in here: https://pubmed.ncbi.nlm.nih.gov/?term=polypharmacy+drug+drug+interaction
  2. Line 313 onward: Better to add more biologically relevant references from PUBMED to showcase the biology relevance of your findings. You can find the references also in the same link here: https://pubmed.ncbi.nlm.nih.gov/?term=polypharmacy+drug+drug+interaction
  3. The formulas no.7-10 are considered common knowledge, so they are not necessary to be showcased in the manuscript
  4. There are some effort to annotate drug adverse effect, such as FAERS database in here: https://www.fda.gov/drugs/questions-and-answers-fdas-adverse-event-reporting-system-faers/fda-adverse-event-reporting-system-faers-public-dashboard . As adverse effect annotation is not really showcased accordingly in this manuscript, and it is an important possible consequences of DDI effects, you should mention it in the discussion and/or conclusion that annotating adverse effect should be your priority in the next venue of this research.

Author Response

Our response: We greatly appreciate the reviewer’s effort to review our manuscript and provide a lot of constructive comments and valuable suggestions for improving our manuscript. We have added relevant references in Sections of Introduction and Interpretability Case Studies to strengthen the biological foundation of the manuscript.

1. Line 28-37: Better add more extensive references on polypharmacy and drug-drug interaction to strengthen the biological foundation.

Our response: We have added relevant references on polypharmacy at the beginning of Section of Introduction.

2. Line 313 onward: Better to add more biologically relevant references from PUBMED to showcase the biological relevance of your findings.

Our response: We have added relevant biological interpretations and references to our findings on page 17 of Section 3.5 Interpretability Case Studies.

3. The formulas no.7-10 are considered common knowledge, so they are not necessary to be showcased in the manuscript

Our response: We have deleted the formulas no.7-10.

4. There are some effort to annotate drug adverse effect, such as FAERS database in here:https://www.fda.gov/drugs/questions-and-answers-fdas-adverse-event-reporting-system-faers/fda-adverse-event-reporting-system-faers-public-dashboard . As adverse effect annotation is not really showcased accordingly in this manuscript, and it is an important possible consequences of DDI effects, you should mention it in the discussion and/or conclusion that annotating adverse effect should be your priority in the next venue of this research.

Our response: According to this suggestion, we selected three interactions between Sildenafil and other Nitrate-based drugs as case studies in Section 3.5. we have added the description of adverse effects between them. In addition, we have added it in Section of Conclusion of our manuscript.

Reviewer 2 Report

The authors present a new perspective of study accounting drug-drug interaction effects. They have exploited very well how drugs can interact and cause side effects to organisms. The way the manuscript was presented was very informative and concise. However, I did not see any  classical simulation or quantum mechanics calculation whereas it will be enrich the quality of the study as well as its support for results. Regarding the presented content, the only point I want to see improved is the conclusion section. In my opinion, they have to show in details for the readers, more about their achievements. The conclusion was very short and not that informative. Therefore, after this modification, the paper should be accepted.

Author Response

Our response: Thank the reviewer for the constructive comments that help our extended manuscript much clearer. According to this suggestion, we have revised the conclusion of our manuscript to include the achievements and more detailed contents.

Round 2

Reviewer 1 Report

The authors have addressed my concerns accordingly. Therefore, I decided to accept this manuscript.